# Description of Baseline Nutrition and Physical Activity Knowledge and Behavior in Acute Stroke/TIA Patients Enrolled in the Health Education on Information Retention and Behavior Change in Stroke (HERBS) Pilot Trial

**DOI:** 10.3390/nu15173761

**Published:** 2023-08-28

**Authors:** Hannah Uhlig-Reche, Diana Ontiveros, Riley Syzdek, Patenne Mathews, Leanne Dalal, Andrea Amaro, Nidhi Wunnava, Zina Housammy, Barrie Schmitt, Anjail Sharrief, Nicole R. Gonzales

**Affiliations:** 1Department of Rehabilitation Medicine, University of Texas Health Science Center at San Antonio, San Antonio, TX 78229, USA; 2Department of Neurology, University of North Carolina School of Medicine, Chapel Hill, NC 27514, USA; 3McGovern Medical School, University of Texas Health Science Center at Houston, Houston, TX 77030, USAleanne.dalal@mhd.com (L.D.); zina.housammy@utsouthwestern.edu (Z.H.); 4School of Public Health, University of Texas Health Science Center at Houston, Houston, TX 77030, USA; 5Department of Internal Medicine, UT Southwestern Medical Center, Dallas, TX 75235, USA; 6Department of Neurology, Neurohospitalist & Stroke Section, University of Colorado School of Medicine, Aurora, CO 80045, USAnicole.r.gonzales@cuanschutz.edu (N.R.G.); 7Department of Neurology, McGovern Medical School, University of Texas Health Science Center at Houston, Houston, TX 77030, USA

**Keywords:** nutrition, physical activity, knowledge, health behavior, self-efficacy, behavioral intent, stage of change, barriers to change, stroke, prevention

## Abstract

Lifestyle modifications after stroke are associated with better risk factor control and lower mortality. The primary objective of this study was to describe the knowledge of American Heart Association (AHA) recommendations for diet and exercise in survivors of stroke and transient ischemic attack (TIA). The secondary objectives were to describe their diet and exercise behaviors, self-efficacy (SE), behavioral intent (BI), stage of change, and barriers to change. Data are described from participants enrolled in a prospective educational intervention in mild stroke/TIA survivors. A multiple-choice questionnaire ascertained knowledge of AHA recommendations for diet and exercise, nutrition and physical activity behavior, SE, BI, stage of change, and barriers to change. Twenty-eight stroke/TIA survivors, with a mean age of 61.7 ± 11.8 years, completed questionnaires during their acute hospitalization. Participants underestimated the recommended intake of fruits, vegetables, whole grains, and participation in aerobic exercise and overestimated the recommended intake of sugar and salt. SE demonstrated a significant positive association with combined behavior scores (r_s_ = 0.36, *p* = 0.043). Greater knowledge of the AHA recommendations was not associated with healthier behavior, greater SE, higher BI, or more advanced stage of change. The gaps between AHA recommendations and stroke/TIA patient knowledge identifies an area for potential intervention in stroke prevention and recovery.

## 1. Introduction

Increasing physical activity, maintaining healthy weight, and adhering to healthy dietary recommendations reduce the risk of cerebrovascular events and lower mortality after stroke [1,2,3,4,5,6]. The American Heart Association (AHA) promotes lifestyle guidelines that support cardiovascular health and disease reduction [5]. Recommendations include high intake of vegetables, fruits, and whole grains and limited intake of sodium, added sugar, and saturated fat. Most Americans do not meet these recommendations; fewer than 25% meet the recommendation for vegetable or fruit intake and approximately 70% exceed the recommended intake of added sugar or saturated fat [7]. A western dietary pattern, such as in the US, is composed of energy-dense and processed foods and has implications for global dietary patterns as westernization expands [8]. This dietary pattern is associated with an increased risk of noncommunicable diseases such as cerebrovascular disease, heart disease, diabetes, and obesity [5].

Complementary to diet, physical activity is essential to leading a healthy lifestyle. Exercise plays a role in energy balance, fitness level, cardiometabolic risk reduction, and optimizing overall health. Self-reported low physical activity frequency has been associated with increased risk of incident stroke [9]. A cross-sectional analysis found lower physical activity levels in US adults living in the Stroke Belt, a region with elevated stroke incidence and mortality, compared to those in non-Stroke Belt states [10]. The AHA recommendations for physical activity are in concordance with the World Health Organization Guidelines and Physical Activity Guidelines for Americans which all recommend that adults perform at least 150 min of moderate-intensity, or 75 min of vigorous-intensity, aerobic exercise a week as well as muscle-strengthening activities at least two days weekly [11,12]. Only 26% of men and 19% of women report sufficient activity to meet the aerobic exercise and muscle-strengthening guidelines in the US, varying by geographic region [6,13]. A recent cohort study of US adults estimated that approximately 110,000 deaths annually could be prevented if moderate-vigorous physical activity was increased by just ten minutes a day [6].

According to the construct of behavioral capability, before a person can perform a behavior successfully they must first possess the knowledge about what actions need to be taken and how to successfully perform them [13,14]. Patients who are counseled by a physician on lifestyle changes are more likely to engage in dietary modification and exercise [15]. While stroke patients identify post-stroke diet management as their primary topic of interest, physicians prioritize topics such as post-stroke medications, rehabilitation, management of post-stroke problems, and medical knowledge about stroke [15,16]. Studies have found that despite the majority of neurologists-in-training believing diet is an important component of stroke prevention, few consistently offer nutritional counseling perhaps due to inadequate training on appropriate dietary counseling [17,18,19,20]. Assessing the baseline knowledge of lifestyle recommendations in stroke patients allows for the identification of gaps, which can inform targeted educational efforts. To our knowledge, no studies have investigated stroke patient knowledge of AHA guidelines for a healthy diet and exercise.

The primary objective of this study was to describe the baseline knowledge of AHA recommendations for nutrition and physical activity in a cohort of stroke and transient ischemic attack (TIA) survivors enrolled in a pilot trial described below. Secondary objectives were to describe their diet and physical activity behaviors, self-efficacy (SE), behavioral intent (BI), stage of change, and barriers to change. Relationships between these domains of interest are explored. The hypothesis was that a minority of participants would have accurate knowledge of AHA recommendations for nutrition and physical activity. We further hypothesized that greater knowledge of the AHA recommendations would be associated with healthier behavior, greater SE, higher BI, and more advanced stage of change.

## 2. Materials and Methods

### 2.1. Study Design

All enrolled participants completed a baseline multiple-choice assessment in the post-stroke acute hospitalization setting, the results of which are reported here. Six domains of interest were assessed: (1) knowledge of AHA recommendations for nutrition and physical activity; (2) nutrition and physical activity behavior; (3) SE; (4) BI; (5) stage of change; and (6) barriers to change. Demographics and clinical characteristics were gathered from the electronic medical record as well as written assessment.

### 2.2. Participants

Participants were recruited from the inpatient service at a Comprehensive Stroke Center in Houston, Texas, to enroll in a pilot trial, Health Education on Information Retention and Behavior Change in Stroke (HERBS). In brief, HERBS is a prospective educational intervention in mild stroke (National Institutes of Health Stroke Scale < 10) and TIA survivors and a cohabitating family member (termed “caregiver”). Inclusion criteria included being diagnosed with acute ischemic or hemorrhagic stroke (NIHSS < 10) or TIA, English-speaking, at least 18 years of age, and having completed at least a sixth-grade level of education. Exclusion criteria included NIHSS ≥ 10, possible malignant stroke etiology, documented dementia/cognitive impairment, living in a nursing home or hospice setting, communication barriers (i.e., aphasia, non-English speaking), and oral intolerance/alternative feeding. The study population comprises a convenience sample of twenty-eight participants.

### 2.3. Evaluations

Information regarding dietary practices was collected using the National Health and Nutrition Examination Survey (NHANES) Dietary Screener Questionnaire (DSQ) [21]. Foods were categorized into four groups: fruits/vegetables, whole grains/beans, sugar, and fat. Free-response questions were excluded as well as question three since cereal intake did not fit into a single scoring category (i.e., whole grain vs. sugar, depending on the type of cereal). Given the small sample size, the scoring procedures were adjusted to create four response groups instead of the original nine and assigned points according to frequency of consumption: 0 points = Never, 1 point = 1–4 times last month, 2 points = 2–6 times per week, 3 points = 1+ times per day.

A fruit/vegetable intake score and a whole grain/bean intake score were calculated by summing the points for the relevant screening questions. Fruit/vegetable intake score was composed of questions 12, 13, 15, 18, 19, and 21, resulting in a maximum possible score of 18 points. Whole grain/bean intake score was comprised of questions 16, 17, 25, and 30, resulting in a maximum possible score of 12 points. Healthy diet score was calculated by summing the points from fruit/vegetable and whole grain/bean intake scores, resulting in a maximum possible score of 30 points. Similarly, sugar intake and fat intake were scored by summing the points scored from the relevant screening questions. Sugar intake score was comprised of questions 8–11 and 26–29, resulting in a maximum possible score of 24 points. Fat intake score included questions 6, 14, 20, and 22–24, resulting in a maximum possible score of 18 points. An unhealthy diet score was calculated by summing the sugar and fat intake scores, resulting in a maximum possible score of 42 points. Higher scores indicate increased frequency of consumption of those food items. For example, high healthy diet score indicates the consumption of high amounts of foods in healthy categories (i.e., fruits/vegetables); high unhealthy dietary score indicates the consumption of high amounts of foods in unhealthy categories (i.e., sugar, fat). Mean diet scores were then converted to approximate daily frequencies [22].

Physical activity practices were collected using questions from the validated Community Healthy Activities Model Program for Seniors questionnaire [23]. A physical activity behavior score was calculated based on frequency of behavior. One point was assigned for each “yes” response to questions regarding the performance of aerobic or strength physical activity, with additional points assigned according to the total duration of weekly exercise (1 point = less than 1 h, 2 points = 1–2.5 h, 3 points = 3+ h). The maximum possible score was 8 points. Higher scores indicate increased physical activity. Combined behavior was calculated by adding healthy diet score with physical activity behavior score then subtracting unhealthy diet score.

There are no validated tools available to assess knowledge of AHA recommendations for nutrition and physical activity or portion sizing. Sixteen multiple-choice questions were developed to assess knowledge comprised of identical language to that used on published educational handouts (Appendix A) [24,25,26]. Questions with one correct answer had 4–5 answer choices. One question had eight choices, of which, respondents were instructed to “check all that apply”. Points were given for correct answers without deductions for incorrect answers. Subdomains were created for nutrition knowledge (14 questions) and physical activity knowledge (2 questions), with greater representation of nutrition knowledge due to the breadth of information available on AHA educational handouts.

The validated Eating Habits Confidence Survey and Exercise Confidence Survey were used to measure SE for diet and physical activity behavior [27,28]. BI was assessed using questions modeled after these two surveys. Both SE and BI were assessed on a five-point Likert Scale (1 = I know I cannot/will not, 5 = I know I can/will). A positive SE/BI is defined as those rated as a 4 or 5; a negative SE/BI is defined as those rated as a 1 or 2, with 3 considered neutral. Five questions assessed SE (Cronbach alpha = 0.76) for a possible total scoring range of 5–25 points. Of those five questions, three pertained to nutrition and two to physical activity. Four questions assessed BI (Cronbach alpha = 0.81), for a possible total scoring range of 4–20 points, of which, three questions assessed nutrition and one assessed physical activity.

One question assessed stage of change in accordance with the transtheoretical model and was scored with higher values indicating a greater readiness for change [29]. The Precontemplation stage reflects those who have not thought about changing their diet/activity level. The Contemplation stage reflects those who reported thinking about changing their diet/activity level. Those planning to change their diet/activity level are reflected by the Preparation stage. The Action stage comprises those who had already made changes to improve their diet/activity level. Those who maintained this improved behavior for the last 6 months are represented by the Maintenance stage. 

One question assessed barriers to change, which had nine answer choices, of which, respondents could select multiple. Response options included: don’t want to, don’t know how (i.e., lack of knowledge), cost/don’t have enough money, don’t have time, don’t have access, weakness/pain, lack of family support, nothing, and other (with write-in option).

### 2.4. Data Analysis

Descriptive statistics were performed and are reported as appropriate, including mean, standard deviation, median, and interquartile range according to Shapiro–Wilk assessed normality. Cronbach’s alpha was calculated for SE and BI questions [30]. With the exception of stages of change and barriers to change*,* all assessment outcomes are continuous variables. The Spearman correlation method was used to evaluate the correlation between the continuous domains of interest, reported as a Spearman correlation coefficient (r_s_). Each domain was broken down into subdomains of nutrition and physical activity and included in correlation analyses. Statistical significance was defined as a *p*-value less than 0.05. Data analyses were performed in SAS software version 9.4. SAS and all other SAS Institute Inc. product or service names are registered trademarks or trademarks of SAS Institute Inc., Cary, NC, USA.

## 3. Results

There were 28 stroke/TIA survivors and 3 caregivers enrolled in the HERBS Trial. Given the particularly small sample of caregivers, only survivor results are reported here. The mean age of participants was 61.7 years, 53.6% are female, and >60% of participants were from racial/ethnic minoritized groups. Demographics and baseline clinical characteristics are shown in Table 1. The associations between knowledge of nutrition and physical activity recommendations with nutrition and physical activity behaviors are shown in Table 2.

### 3.1. Nutrition Knowledge

The mean (SD) nutrition knowledge score was 6.0 (2.0) out of 14.0 (42.9% correct). The maximum score achieved was 9 points (64.3% correct). Nutrition knowledge demonstrated a positive association with SE (r_s_ = 0.45, *p* = 0.064), but no statistically significant associations were found with the domains of interest.

Data for individual questions are available in Appendix A, but interesting results are described here. The AHA recommendation to consume at least five servings of vegetables daily was correctly identified by 32.1% of respondents. A total of 63% of respondents incorrectly identified a lesser amount. A total of 37% correctly identified that the AHA recommends consuming at least 4 servings of fruit daily and 63% identified a lesser amount. The AHA recommendation for whole grain consumption of 3–6 servings per day was correctly identified by 11.1% of respondents, with 81.5% incorrectly identifying a lesser amount. A total of 64% correctly identified the AHA recommendation to limit daily added sugar to 100 calories a day (6 teaspoons) for women and 150 calories (9 teaspoons) for men, with the remaining 36% incorrectly identifying a greater amount. The majority (62.5%) correctly identified sugar-sweetened beverages as the greatest source of added sugar in the American diet. Approximately 30% of respondents correctly identified the AHA recommendation to limit sodium to 1,500 milligrams per day, but a greater amount was incorrectly identified by 15.4% of respondents. For portion sizing, 51.9% correctly identified that one cup of food is approximately equal in size to a fist or a baseball and 50% correctly identified that three ounces of meat is approximately equal in size to a palm or deck of cards. 

### 3.2. Nutrition Behavior

Mean (SD) healthy diet score was 10.4 (3.9) out of 30 (34.7%). Mean (SD) fruit/vegetable intake score was 7.1 (2.8) out of 18 (39.4%). Mean (SD) whole grain/bean intake score was 3.3 (2.0) out of 12 (27.5%). Mean (SD) unhealthy diet score was 14.2 (5.3) out of 42 (33.8%). Mean (SD) fat intake score was 5.9 (2.8) out of 18 (32.8%). Mean (SD) sugar intake score was 8.7 (3.6) out of 24 (36.3%). See Figure 1 for approximate daily frequency of intake. The significant associations between healthy diet with whole grains and fruits and vegetables (Table 2) is reflective of the scoring procedure, as these food group scores comprise the healthy diet score.

Unhealthy diet score demonstrated a positive trend with fruit/vegetable intake score (r_s_ = 0.36, *p* = 0.056). Fat intake score demonstrated a statistically significant positive association with added sugar intake score (r_s_ = 0.34, *p* = 0.034) and positive trend with fruit/vegetable (r_s_ = 0.36, *p* = 0.076) score. The significant associations between unhealthy diet with added sugars and fats (Table 2) are reflective of the scoring procedure, as these food group scores comprise the unhealthy diet score. 

Baseline dietary intake frequency of fruits/vegetables, whole grains/beans, sugar, and fat per day (*n* = 28) are presented in Figure 1. We are unable to report quantity of servings due to lack of portion size information.

### 3.3. Physical Activity Knowledge

The median (IQR) physical activity knowledge score was 1 (0.0, 1.0) out of 2 points. Briefly, 29.6% correctly identified 150 min as the AHA recommendation for minimum time spent performing moderate-intensity aerobic physical activity per week. A lesser amount was incorrectly identified by 63% of respondents. The AHA recommendation for strength training at least twice per week was correctly identified by 29.6% of respondents and 18.5% incorrectly identified a lesser amount. There were no statistically significant associations between physical activity knowledge and the other domains of interest.

### 3.4. Physical Activity Behavior

The median (IQR) physical activity behavior score was 1.5 (0.0, 4.0) out of 8.0 (18.75%) (Figure 2). Nearly 43% reported doing aerobic exercise during a typical week in the last four weeks, with a median (IQR) weekly frequency of 3 (2, 6). Approximately 42% of those reported participating in greater than 2.5 h of exercise per week and another 42% participated in less than one hour per week. Twenty-five percent of respondents reported doing strength-training exercise during a typical week in the last four weeks, with a median (IQR) weekly frequency of 3 (2.5, 6.5). Approximately 43% of respondents reported performing strength training for less than 1 h, 43% for 1–2.5 h, and 14% for 3–4.5 h per week. There were no statistically significant associations between physical activity behavior and the other domains of interest.

In Figure 2, baseline knowledge, behavior, self-efficacy, and behavioral intent are each displayed as a percentage of maximum points possible. Behavior is displayed as healthy diet, unhealthy diet, and physical activity. According to normality, we display mean scores for knowledge and behavior and median scores for self-efficacy and behavioral intent.

### 3.5. Combined Knowledge

Combined knowledge comprised scores for both nutrition and physical activity knowledge, with a maximum possible score of 16 points. Mean (SD) score was 6.5 (2.2) out of 16, or 40.6% correct. The maximum score achieved was 9.5 out of 16, or 59.4%. There were no statistically significant associations between combined knowledge and the other domains of interest.

### 3.6. Combined Behavior

The median (IQR) combined behavior score was −1.5 (−6.0, 3.0). Greater behavior scores were significantly associated with increased SE (r_s_ = 0.36, *p* = 0.043), as shown in Table 3.

### 3.7. Self-Efficacy

Median (IQR) total SE was 18 (15.5, 21.0) out of 25 points (72%) (Figure 2). Highest SE was reported for eating poultry/fish instead of red meat at dinner at 75% with positive SE (rated as a 4 or 5). Lowest SE was reported for ability to avoid adding salt at the table, with 32.1% reporting a negative SE (rated as a 1 or 2). See Figure 3a. Self-efficacy was positively associated with combined behavior (r_s_ = 0.36, *p* = 0.043).

### 3.8. Behavior Intent

Median (IQR) total SE was 18.5 (14.0, 20.0) out of 20 points (92.5%) (Figure 2). Highest BI was reported for eating more fruits and vegetables with 85.7% indicating a 4 or 5 on the Likert scale. See Figure 3b.

### 3.9. Stage of Change

Figure 4 presents the proportion of 26 respondents identifying each stage of change: precontemplation (3.9%), contemplation (19.2%), preparation (38.5%), action (11.5%), maintenance (11.5%), relapse (15.4%).

### 3.10. Barriers to Change

Nearly 30% of respondents reported barriers to change as “none”. The most frequently identified barrier to change was lack of knowledge (29.6%) followed by lack of access (25.9%). Figure 5 displays the percentage of respondents identifying any relevant baseline barriers to change (*n* = 27). Respondents could select multiple barriers. Lack of knowledge (“don’t know how”) was the most identified barrier, with an equal number reporting no barriers. Lack of family support was the least often identified barrier to change. No “other” barriers were identified or written in.

## 4. Discussion

In this study of stroke/TIA survivors, results showed poor knowledge and performance of AHA recommendations for diet and physical activity, but high SE and BI. Fat intake score demonstrated a positive association with sugar intake score. SE demonstrated a significant positive association with higher combined behavior scores. Most participants were at least thinking about making healthy lifestyle changes. A commonly cited barrier to change was lack of knowledge. 

This is the first study to assess acute stroke survivor knowledge of AHA diet and physical activity recommendations. Results show poor knowledge of AHA guidelines for both nutrition and physical activity. Most participants underestimated the AHA recommendations for intake of fruits, vegetables, whole grains, and participation in aerobic exercise. Per the construct of behavioral capability, this lack of knowledge may contribute to the measured deficits in reaching behavioral goals. This is supported by a study in underserved African American middle-aged and older adults with hypertension, which found that those who had higher level of hypertension knowledge were more likely to adhere to lifestyle recommendations and medication regimen [31]. Knowledge of nutrition recommendations was nearly positively associated with SE, suggesting that health education may increase behavioral confidence. While results in this study did not reveal a significant association between knowledge and behavior, the study is limited by the small sample size. This unique setting in which participants are recovering from acute cerebrovascular insult may have impacted their ability to recall information, thereby affecting their scores. If patients at risk of stroke had assessments conducted prior to neurologic injury, perhaps their knowledge or recall of behavior would be different. It is also possible that one’s lifestyle practices are unconcordant with knowledge. For example, strong taste preferences may supersede dietary knowledge. Findings may still be clinically relevant and additional studies should be conducted.

Few participants reported healthy diet and exercise practices, aligning with that of the general US population. Most Americans consume less than the recommended amounts of fruits and vegetables, adequate total grains, and have an intake at or above the recommended limits for added sugar and saturated fat [7]. Adherence to healthy lifestyle practices plays an importance role in promoting cerebrovascular health due to its effect on vascular risk factors including management of blood glucose, blood lipids, and blood pressure. The stroke prevention treatment plan is incomplete without addressing healthy lifestyle behavior.

It is important to recognize that the DSQ does not ascertain portion sizes, only frequency of consumption. As such, it is difficult to quantify intake by servings per food group. Because portion sizes have increased over the years, but not uniformly so, it is likely that the frequency of fat and sugar intake translates to a greater number of servings consumed compared to fruits/vegetables [32]. For example, restaurant, fast food, and convenience foods offer larger portions while a singular fruit (i.e., apple) remains relatively consistent [33]. Despite this limitation in dietary measurement, it was determined that respondents consumed fruits/vegetables relatively infrequently. A recent study in patients with recent stroke similarly demonstrated baseline mean intake of fruits/vegetables per day to be below the AHA recommendations but did not report intake of other food groups [34]. The positive association between fat and sugar intake scores suggests that those who make poor dietary choices in one category also do so in another.

A minority of respondents reported performing aerobic exercise and, of those who did, less than half met the AHA recommendation for total weekly duration. Even fewer respondents reported strength training, but most of those who did met the recommended weekly frequency. A recent study in post-stroke patients defined “health enhancing physical activity” as 30 min for 3 days a week and found that only 9.8% of their study population met this criterion [34,35]. Although the authors used a different self-reported measurement tool, our findings indicate that stroke patients in this study led relatively inactive lifestyles. 

SE is one of the strongest predictors of health behaviors such as physical activity and fruit/vegetable intake [36]. This study revealed high SE for health behavior in an acute stroke/TIA population, similar to what has been reported in middle-aged or older adults across Texas and elsewhere in the US [37,38]. Highest SE was reported for eating poultry/fish instead of red meat at dinner, so this may be a reasonable goal for initial dietary improvement. Lowest SE was reported for ability to avoid adding salt at the table so employing motivational interviewing to explore barriers to this change may be useful. Greater nutrition knowledge was insignificantly associated with higher SE, suggesting that educating stroke patients about healthy dietary recommendations may increase their confidence for performing the desired behavior. The positive association between combined behavior and SE is aligned with the theory of SE; those who have already accomplished performing healthy behavior will feel more confident in their capacity to perform healthy behaviors. Research has shown that SE is associated with more advanced stage of change regarding physical activity [39].

According to the Theory of Planned Behavior, the stronger one’s intention to perform a behavior, the more likely one will perform that behavior [40]. Greater behavioral intention has predicted healthy eating and exercise behaviors [41,42]. While there was not an association between BI and reported baseline behavior, high BI suggests follow-up behavior may demonstrate improvement.

The most prevalent stages of change in this stroke/TIA population were the planning and contemplation stages. This is similar to what was reported in a study of patients with type 2 diabetes [43]. Like stage of change, “readiness to change” physical activity and dietary behavior in the next six months has been reported by a majority of stroke survivors [34]. It may be that the stroke/TIA serves as a cue to action for these patients, motivating them to want to change their lifestyle. When nutrition education is tailored to one’s stage of change, it may improve dietary behavior [44,45,46]. Tactfully navigating discussions about lifestyle change with stroke/TIA survivors based on the relevant stage of change can be a meaningful approach [47]. For example, those in the contemplation stage should be encouraged to explore both the negative and positive aspects of improved diet and exercise. Individuals in the preparation stage should be provided with more concrete guidance on behavior change and should identify supports and barriers to change [48].

Nearly one-third of respondents identified barriers to change as “none”, which suggests that they may be able to implement improved lifestyle changes easier. The most identified barrier to change in this study was lack of knowledge or “don’t know how”. Similarly, knowledge was found to be a significant barrier to adherence to a healthy diet pattern in Americans, particularly those in the Stroke Belt [49]. Lack of cooking skills has been identified as a significant barrier to meeting fruit/vegetable intake recommendations in women, but not in men [50]. Lacking knowledge/skills to properly exercise has been reported in other populations but has been inconsistently reported in stroke survivors [50,51,52,53,54,55,56]. It is important to note that this study assessed stages of change in the acute post-stroke setting, so patients may not yet recognize new barriers they may experience in recovery (i.e., functional limitations). The lack of knowledge reported in this setting is indicative of a barrier to lifestyle change that existed before stroke/TIA, presenting a barrier targetable in prevention strategies. 

By assessing stroke/TIA survivors in the acute hospitalization setting, the information that was gathered is relevant to their cerebrovascular risk factors. This study is unique as it is the first, to the best of our knowledge, to assess knowledge of AHA recommendations for healthy diet and physical activity in survivors of stroke/TIA. The lack of validated tools to assess this knowledge is a limitation. Multiple-choice questions may result in correct answers by chance alone (20–25%, depending on number of answer choices). 

The findings of this study have implications for both primary and secondary stroke prevention. Pre-stroke lifestyle information was gathered by asking about recent diet and exercise, providing insight into targets for primary prevention. The DSQ utilized here is limited by recall bias and lack of portion size information. Despite this, multiple important food groups are assessed to investigate stroke survivors’ dietary behavior. The physical activity behavior is similarly limited by recall bias and the vigorousness of exercise is unknown. However, assessing both diet and exercise behaviors provides valuable insight which can be used to guide prevention strategies. The barriers to change reported here reflect barriers that prevented lifestyle changes before stroke, again important to consider in primary prevention. Stroke/TIA may serve as a call to action thereby stimulating patients to consider lifestyle improvements, increasing their SE and BI to do so.

The results of this study may not be generalizable due to the small sample size and convenience sample at a single site in the southwestern United States. Future studies should utilize a larger sample size across multiple recruitment sites that vary in geographic area. Despite the study population being all English-speaking, the group is fairly representative of this particular patient population. However, ethnicity was unable to be determined for several participants due to limitations in the electronic medical record. Results would be more generalizable if non-English speaking patients were included. Future studies should investigate these domains in a larger, more diverse population of stroke/TIA survivors.

## 5. Conclusions

In this cohort of individuals with stroke/TIA survivors, results showed poor knowledge and performance of AHA recommendations for diet and physical activity, but high SE and BI. Fat intake score demonstrated a positive association with sugar intake score. Most participants were at least thinking about making healthy lifestyle changes. A commonly cited barrier to change was lack of knowledge. SE demonstrated a significant positive associated with higher combined behavior scores. Overall, the results suggest a need to educate stroke/TIA survivors in the US on AHA recommendations for healthy diet and physical activity. By overcoming this gap in knowledge, those with cerebrovascular risk factors may feel empowered to improve their nutrition and exercise behaviors in accordance with AHA guidelines and thereby prevent a secondary cerebrovascular event. The information ascertained in this study can be used to inform future educational efforts.

## Figures and Tables

**Figure 1 nutrients-15-03761-f001:**
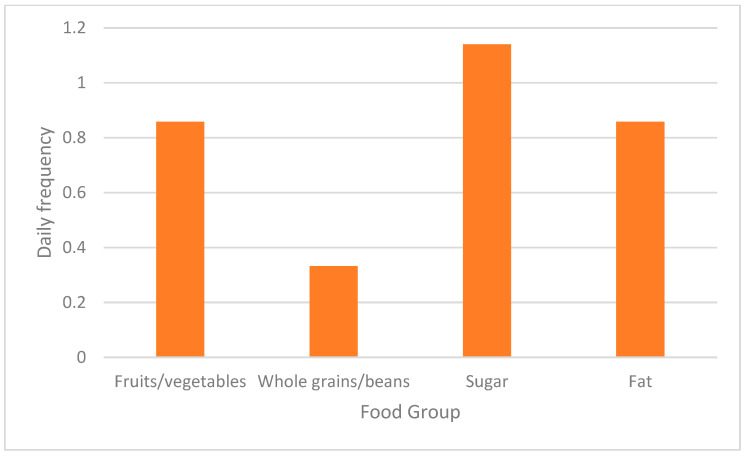
Estimated Frequency of Food Group Intake per Day.

**Figure 2 nutrients-15-03761-f002:**
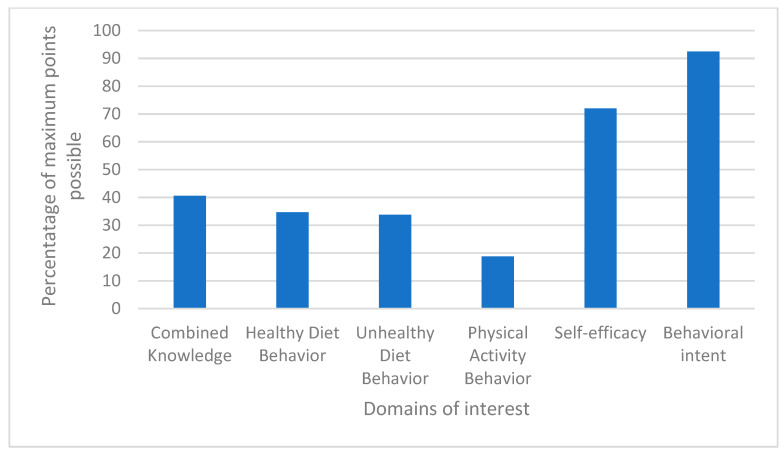
Baseline Participant Domain Data.

**Figure 3 nutrients-15-03761-f003:**
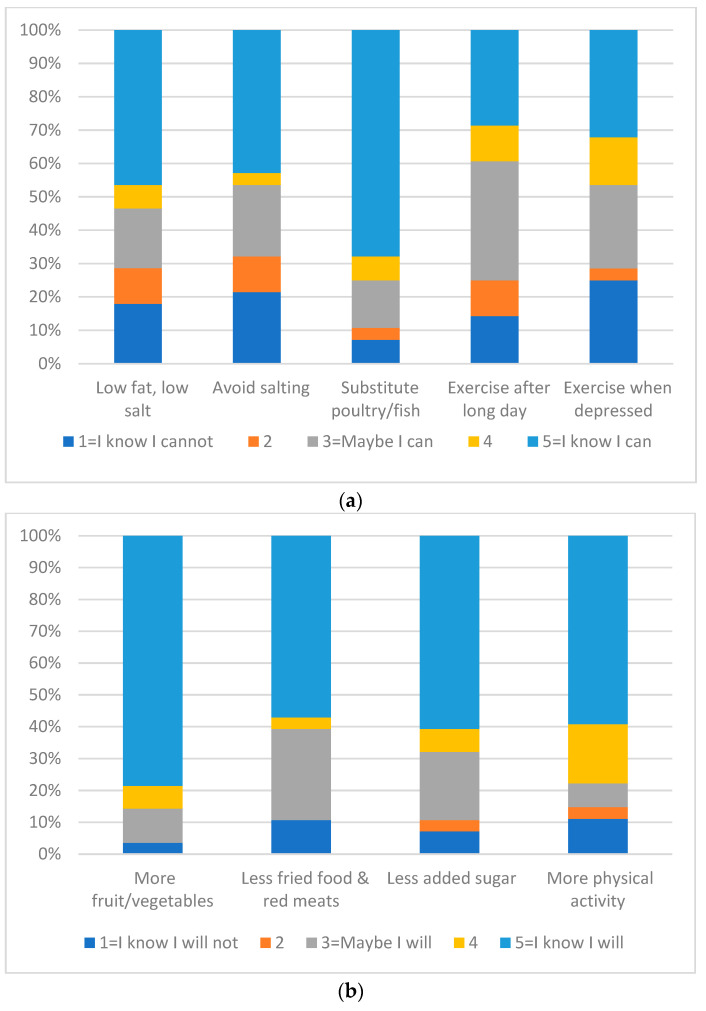
Self-efficacy and behavioral intent. (**a**) Self-efficacy responses of 28 respondents on a five-point Likert Scale (1 = I know I cannot, 5 = I know I can). (**b**) Behavioral Intent responses on a five-point Likert Scale (1 = I know I will not, 5 = I know I will). N = 28 except for physical activity, where *n* = 27.

**Figure 4 nutrients-15-03761-f004:**
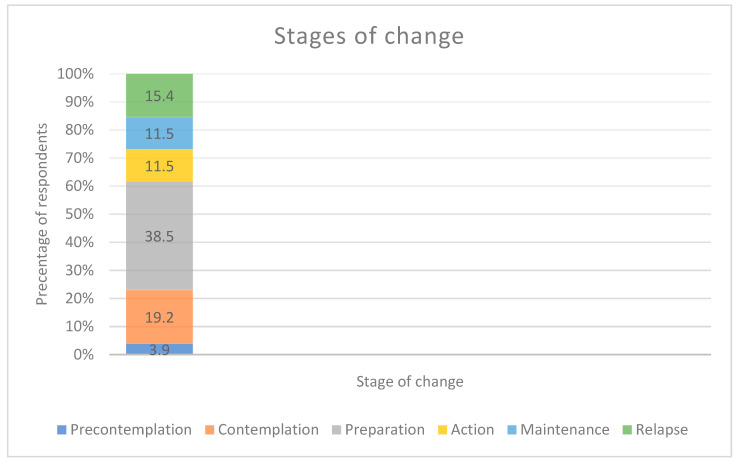
Stages of Change.

**Figure 5 nutrients-15-03761-f005:**
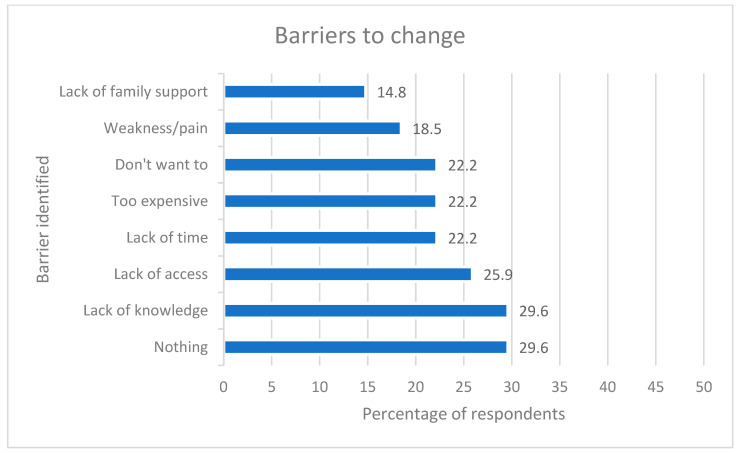
Barriers to change.

**Table 1 nutrients-15-03761-t001:** Participant Demographics and Baseline Clinical Characteristics.

Race/Ethnicity, *n* (%)	
White	3 (10.7)
Black	14 (50.0)
Hispanic	4 (14.3)
Other	1 (3.6)
Unable to determine	6 (21.4)
Age, mean (SD) (years)	61.7 (11.8)
Sex, *n* (%)	
Male	13(46.4)
Highest level of education completed, *n* (%)	
<12 years of education	8 (28.6)
High school graduate or GED	8 (28.6)
>12 years of education	12 (42.9)
Marital Status, *n* (%)	
Married or domestic partnership	16 (57.1)
Without a partner (divorced, widowed, single)	12 (42.9)
Lives alone, *n* (%)	
Yes	8 (28.6)
Annual Income, *n* (%)	
≤49,999	15 (53.6)
>$49,999	10 (35.7)
Decline to answer NIHSS, median (IQR)	3 (10.7) 4.0 (3.0, 7.0)
Labs on Admission	
Creatinine, median (IQR) (mg/dL)	1.0 (0.9, 1.2)
Cholesterol, total, median (IQR) (mg/dL)	170.0 (133.0, 197.0)
Cholesterol, LDL, mean (SD) (mg/dL)	99.9 (41.9)
Cholesterol, HDL, mean (SD) (mg/dL)	50.9 (13.2)
Triglycerides, median (IQR) (mg/dL)	106.0 (67.0, 130.0)
Glucose, median (IQR) (mg/dL)	105.5 (94.0, 179.0)
Glycated Hemoglobin, median (IQR) (%)	5.9 (5.4, 7.5)
Weight, median (IQR) (kg)	84.1 (77.3, 102.3)
BMI, median (IQR) (kg/m^2^)	31.0 (25.1, 38.9)
Systolic BP on admission, mean (SD), mm Hg	144.7 (23.8)
Diastolic BP on admission, mean (SD), mm Hg	82.2 (18.0)
tPA administered, *n* (%)	
Yes	7 (25.0)
Intra-arterial therapy, *n* (%)	
Yes	4 (14.3)
Discharge Diagnosis, *n* (%)	
TIA	2 (7.1)
Stroke	26 (92.9)
Hospital Length of Stay (days), median (IQR)	5.0 (2.0, 8.0)
Discharge Location, *n* (%)	
Home	18 (64.3)
Other	10 (35.7)
Stroke Etiology *, *n* (%)	
Small artery occlusion	7 (25.0)
Large artery occlusion	5 (17.9)
Cardioembolic	5 (17.9)
Undetermined/cryptogenic	11 (39.3)
Medications prior to admission, *n* (%)	
Antiplatelet	13 (46.4)
Anticoagulant	0 (0)
Beta blocker	8 (28.6)
ACEi/ARB	8 (28.6)
Other anti-hypertensive	12 (42.9)
Diabetic agents	6 (21.4)
Statin	11 (39.3)
Reported medication compliance, *n* (%)	
No	2 (7.1)
Yes	5 (17.9)
Unable to determine ^ᵞ^	15 (53.6)
Not applicable	6 (21.4)

Abbreviations: ACEi/ARB—Angiotensin-Converting Enzyme Inhibitor/Angiotensin II Receptor Blocker; BMI—body mass index; BP—blood pressure; HDL—high density lipoprotein; IQR—interquartile range; LDL—low density lipoprotein; *n*—sample size; NIHSS—National Institutes of Health Stroke Scale; SD—standard deviation; tPA—tissue Plasminogen Activator. * All participants experienced ischemic stroke. ^ᵞ^ The medical record lacked explicit documentation regarding patient compliance or noncompliance.

**Table 2 nutrients-15-03761-t002:** Associations between knowledge of nutrition and physical activity recommendations with nutrition and physical activity behaviors.

	Nutrition Knowledge r_s_ (*p* Value)	Healthy Diet r_s_ (*p* Value)	Healthy Diet, Fruit and Vegetable r_s_ (*p* Value)	Healthy Diet, Whole Grains r_s_ (*p* Value)	Unhealthy Diet r_s_ (*p* Value)	Unhealthy Diet, Fat r_s_ (*p* Value)	Unhealthy Diet, Added Sugars r_s_ (*p* Value)	Physical Activity Knowledge r_s_ (*p* Value)	Physical Activity Behavior r_s_ (*p* Value)
Nutrition knowledge	N.A.	0.189 (0.335)	0.255 (0.190)	−0.064 (0.746)	−0.120 (0.543)	−0.063 (0.749)	−0.141 (0.474)	0.088 (0.663)	0.080 (0.686)
Healthy diet	0.189 (0.335)	N.A.	**0.909** **(<0.001)**	**0.620** **(<0.001)**	0.274 (0.158)	0.265 (0.172)	0.127 (0.521)	−0.330 (0.092)	0.256 (0.188)
Healthy diet, fruit and vegetables	0.255 (0.190)	**0.909** **(<0.001)**	N.A.	0.265 (0.173)	0.362 (0.059)	0.340 (0.076)	0.216 (0.269)	−0.245 (0.218)	0.239 (0.221)
Healthy diet, whole grains	−0.064 (0.746)	**0.620** **(<0.001)**	0.265 (0.173)	N.A.	0.017 (0.933)	−0.0001 (0.100)	−0.007 (0.972)	−0.327 (0.096)	0.239 (0.221)
Unhealthy diet	−0.120 (0.543)	0.274 (0.158)	0.362 (0.059)	0.017 (0.933)	N.A.	**0.758** **(<0.001)**	**0.851** **(<0.001)**	0.133 (0.508)	−0.185 (0.346)
Unhealthy diet, fat	−0.0633 (0.749)	0.265 (0.172)	0.341 (0.076)	−0.0001 (0.100)	**0.758** **(<0.001)**	N.A.	**0.393** **(0.039)**	0.231 (0.246)	−0.103 (0.604)
Unhealthy diet, added sugars	−0.141 (0.474)	0.127 (0.521)	0.216 (0.269)	−0.007 (0.972)	**0.851** **(<0.001)**	**0.393** **(0.039)**	N.A.	0.101 (0.617)	−0.058 (0.770)
Physical activity knowledge	0.088 (0.663)	−0.330 (0.092)	−0.245 (0.218)	−0.189 (0.336)	0.133 (0.508)	0.231 (0.246)	0.101 (0.617)	N.A.	0.091 (0.653)
Physical Activity behavior	0.799 (0.686)	0.256 (0.188)	0.239 (0.221)	0.170 (0.387)	−0.185 (0.346)	−0.105 (0.604)	−0.0579 (0.770)	0.091 (0.653)	N.A.

N.A.—Non-applicable; r_s_—Spearman Correlation Coefficient. Bold text identifies statistical significance.

**Table 3 nutrients-15-03761-t003:** Association of combined knowledge and combined behavior with self-efficacy and behavioral intent.

	Combined Knowledge r_s_ (*p* Value)	Combined Behavior r_s_ (*p* Value)	Self-Efficacy r_s_ (*p* Value)	Behavioral Intent r_s_ (*p* Value)
Combined knowledge	N.A.	−0.010 (0.961)	0.210 (0.283)	−0.012 (0.954)
Combined behavior	−0.010 (0.961)	N.A.	**0.385** **(0.043)**	0.229 (0.242)
Self-efficacy	0.210 (0.283)	**0.385** **(0.043)**	N.A.	0.284 (0.143)
Behavioral intent	−0.012 (0.954)	0.229 (0.242)	0.284 (0.143)	N.A.

N.A.: Non-applicable; r_s:_ Spearman Correlation Coefficient. Bold text identifies statistical significance.

## Data Availability

Pertinent data is contained within the article. Additional data is available on request from the corresponding author.

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
