# Peer review of "Description of Baseline Nutrition and Physical Activity Knowledge and Behavior in Acute Stroke/TIA Patients Enrolled in the Health Education on Information Retention and Behavior Change in Stroke (HERBS) Pilot Trial"

_nutrients, 2023, doi:10.3390/nu15173761_

Round 1

Reviewer 1 Report

The study was essentially a needs assessment for a hospital about the knowledge and such for post-stroke care to develop a program. There were some areas to consider for further strengthening this manuscript:

Abstract:

For the background information, it was mentioned that few stroke survivors receive information about diet and exercise. Please clarify if this is referring to the US, a specific country or globally.

Introduction:

The introduction is quite short and can provide more explanation and clarity to the context as it appears to be written for the US audience when in fact Nutrients is an international journal that reaches a broad base of readership in various countries. For example, there could be a paragraph just focused on the eating habits/behaviors of those who had a stroke or cardiac event or current consumption patterns. In the same token, the type of physical activity that is currently being done and what needs to be done to improve cardiac health.

In the first sentence there is mention of increasing physical activity. While there are statistics to indicate that many people (again depending on the country) are not obtaining the adequate amount of physical activity, there may need to be clarification on what is meant by increasing. Meaning if one does 5 minutes of walking daily, would increasing to 10 minutes be sufficient? Please indicate.

Regarding the second paragraph, it appears that only the physician would communicate with the patient about stroke care. However, depending on the hospital, several have a core team for stroke in which a dietitian, exercise physiologist, social worker, etc work with the patient prior to their discharge to set them up with a cardiac rehab program. Therefore, is the focus of this study then to identify the information not provided to the patients to create educational programs for physicians? Or to enhance the team-based care approach when someone is recovering from a stroke? Please clarify. Also, there may be studies that have been done about dietitians or nutrition educators providing educational programs to those who had a stroke or other cardiac event to identify the success.

As the focus for this study was solely on the AHA guidelines, would indicate if physicians and others must use these guidelines for communicating information with a individual who had a stroke.

Methods:

For the AHA questionnaire, expand on how the validation occurred prior to using it for the study.

For the stage of change information, please provide the reference to these stages and the instrument that was used for this as there are some out there to assess stage of change and readiness.

Include the sample size calculation for this pilot.

Results:

From the demographics and results from the nutrition knowledge, was there a question pertaining to past nutrition education and if they did obtain this information from who? Please clarify.

Discussion:

May consider a non-grouping of the topics to allow for better connection for each topic. For example, there was an association between nutrition knowledge and self-efficacy yet the discussion was focused on the knowledge and then on self-efficacy separate as opposed to interconnected.

Reviewer 2 Report

The article is original and highlights an excellent health issue. This article may contribute to the counseling of patients after a cardiovascular event.

It is very well scientifically grounded, and the introduction objectively addresses the research problem.

Despite the 48 bibliographical references, only 11 are from the last five years.

In the materials and methods section, it is missing to describe the size of the study population and mention that it is a convenience sample population. The evaluation methods used to construct the variables are adequate. The stats used are also good. Due to the small sample size, I suggest reducing the number of population characteristic categories (Table 1). For example, marital status: with or without a partner; Annual income: <$49,999 and >=-$49,999. Add the percentage values in Figure 4. The results shown in Figure 5 are not accurate. Looks like the data label is missing. As this is a convenience sample, the authors should add to the conclusions that the results found refer specifically to this studied population. Therefore, it may not apply to other contexts.
